# Photo-induced semimetallic states realised in electron–hole coupled insulators

Kozo Okazaki [1], Yu Ogawa[1], Takeshi Suzuki[1], Takashi Yamamoto[1], Takashi Someya[1], Shoya Michimae[1], Mari Watanabe[1], Yangfan Lu [2], Minoru Nohara[3], Hidenori Takagi [2,4], Naoyuki Katayama[5], Hiroshi Sawa[5], Masami Fujisawa[1], Teruto Kanai[1], Nobuhisa Ishii[1], Jiro Itatani [1], Takashi Mizokawa [6] & Shik Shin [1]

Using light to manipulate materials into desired states is one of the goals in condensed matter physics, since light control can provide ultrafast and environmentally friendly photonics devices. However, it is generally difficult to realise a photo-induced phase which is not merely a higher entropy phase corresponding to a high-temperature phase at equilibrium. Here, we report realisation of photo-induced insulator-to-metal transitions in $Ta_2Ni(Se_{1-x}S_x)_5$ including the excitonic insulator phase using time- and angle-resolved photoemission spectroscopy. From the dynamic properties of the system, we determine that screening of excitonic correlations plays a key role in the timescale of the transition to the metallic phase, which supports the existence of an excitonic insulator phase at equilibrium. The non-equilibrium metallic state observed unexpectedly in the direct-gap excitonic insulator opens up a new avenue to optical band engineering in electron–hole coupled systems.

[1] Institute for Solid State Physics, University of Tokyo, Kashiwa, Chiba 277-8581, Japan. [2] Department of Physics, University of Tokyo, Bunkyo-ku, Tokyo 113-0033, Japan. [3] Research Institute for Interdisciplinary Science, Okayama University, Okayama 700-8530, Japan. [4] Max Planck Institute for Solid State Research, Heisenbergstrsse 1, 70569 Stuttgart, Germany. [5] Department of Applied Physics, Nagoya University, Nagoya 464-8603, Japan. [6] School of Advanced Science and Engineering, Waseda University, Shinjukuku, Tokyo 1698555, Japan. Correspondence and requests for materials should be addressed to K.O. (email: okazaki@issp.u-tokyo.ac.jp) or to S.S. (email: shin@issp.u-tokyo.ac.jp)

I n semimetals or small-gap semiconductors, valence-band holes and conduction-band electrons may form bound states or excitons via weakly screened Coulomb interaction. The excitons condensate in a Bardeen–Cooper-Schrieffer or Bose–Einstein condensation (BEC) manner, depending on whether the electron–hole coupling is weak or strong, and such a ground state is theoretically predicted as an excitonic insulator[1]. One of the prototypical candidates of excitonic insulators is $1T$-TiSe₂, which shows a charge–density–wave (CDW) transition accompanying a $2 \times 2 \times 2$ structural distortion at ~202 K[2,3]. Figure 1a illustrates a canonical phase diagram of excitonic insulators, and $1T$-TiSe₂ indeed exhibits such a phase diagram. One of the most plausible evidence that $1T$-TiSe₂ is an excitonic insulator has been reported by Hellmann et al.[4]. They have classified several CDW insulators by their dominant interactions to Mott, excitonic, and Peierls insulators based on their melting times of electronic order parameters by time- and angle-resolved photoemission spectroscopy. In addition, a recent electron energy loss spectroscopy study on $1T$-TiSe₂ has reported an electronic collective mode coupled to phonons expected for an excitonic insulator[5]. However, since $1T$-TiSe₂ has indirect-type electron and hole bands (the valence band maximum and the conduction-band minimum are located at different positions in the Brillouin zone), and its excitonic condensation is inevitably accompanied by a band folding with a finite wave vector $q$ like a Peierls insulator, it is still difficult to exclude contribution of electron lattice interaction. Also the indirect gap is not favourable for optical control of electrons and holes for future application.

On the other hand, Ta₂NiSe₅, which has been supposed to be a unique candidate of excitonic insulators in the strong coupling (BEC) regime, has a quasi one-dimensional structure composed of layers of Ni single chains and Ta double chains along the $a$-axis

(Fig. 1b), and each layer is stacked by van der Waals interactions[6,7]. Hybridised Ni $3d$ and Se $4p$ orbitals mainly compose the valence band near the Fermi level ($E_F$), whereas Ta $5d$ orbitals primarily form the doubly degenerate conduction bands (Fig. 1c)[6,7]. This degeneracy can be partially lifted by the finite hybridisation between the two Ta chains. When the temperature decreases, Ta₂NiSe₅ undergoes a semiconductor-to-insulator transition at 328 K, accompanied by a structural distortion from orthorhombic to monoclinic symmetry[6–8]. According to the angle-resolved photoemission spectroscopy (ARPES) measurements in the equilibrium state, it has been found that the top portion of the valence band remarkably becomes flat below the transition temperature. This has been considered as evidence for the spontaneous formation of excitons between the Ta $5d$ electron bands and the Ni $3d$-Se $4p$ hybridised hole bands and thus, a phase transition to an excitonic insulator in the strong coupling regime[9–12]. Recently, Lu et al., have established the phase diagram of Ta₂Ni(Se$_{1-x}$S$_x$)₅, which covers the excitonic insulator phase and the band insulator phase as a function of $x$[13]. However, more direct evidence for the excitonic insulating phase is still lacking so far. If the similar behaviour of the pump-fluence dependence to the photo-excitation to $1T$-TiSe₂ is observed also in Ta₂NiSe₅, it will strengthen that the behaviour can be regarded as the evidence for the excitonic insulating phase.

For the present study, we have performed the measurements of time- and angle-resolved photoemission spectroscopy (TARPES) to obtain such evidence that Ta₂NiSe₅ is actually an excitonic insulator from its pump-fluence dependence of the photo-excitation dynamics. Whereas several time-resolved studies on Ta₂NiSe₅ have been reported so far[14–17], the present study is quite unique in that we employ a pump laser with shorter pulse duration (~30 fs), and extreme ultraviolet laser from high harmonic generation for probe pulses. Whereas we have employed 1 kHz for the repetition rate of the laser in order to generate higher order harmonics, this makes higher pump fluence available compared to higher repetition rate with the same average power of pump pulses. This would be the reason why we obtained the rather different TARPES results from Mor et al.[15], which have revealed the band gap narrowing and the enhanced excitonic coupling in Ta₂NiSe₅ by photo-excitation. We do observe that slightly S-substituted Ta₂NiSe₅ shows a characteristic pump-fluence dependence in its excitation dynamics, whereas Ta₂NiS₅, which had been considered as an ordinary band insulator, shows no dependence. Furthermore, quite unexpectedly, we find a non-equilibrium metallic phase as a photo-excited state of slightly S-substituted Ta₂NiSe₅ as well as Ta₂NiS₅. Whereas our results strongly suggest that Ta₂NiSe₅ is an excitonic insulator, our observation of non-equilibrium metallic phase in both of slightly S-substituted Ta₂NiSe₅ and Ta₂NiS₅ may require reconsidering that Ta₂NiS₅ is not an ordinary band insulator. We propose the importance of the electron correlation effect for the insulating ground state of Ta₂NiS₅. In addition, our findings serve a new pathway to phase control of materials including excitonic insulators by light.

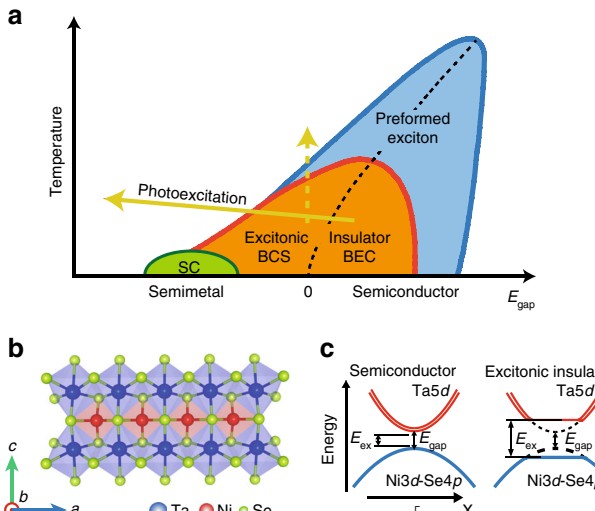

**Fig. 1** Crystal and electronic structure of Ta₂NiSe₅ and phase diagram of excitonic insulators. **a** Phase diagram of excitonic insulators. The solid arrow indicates the pathway for the photo-induced phase transition proposed in this study and the dashed arrow represents the effect of increasing the temperature. **b** Crystal structure of Ta₂NiSe₅. A quasi one-dimensional structure is formed by a single Ni chain and two Ta chains along the $a$-axis. **c** Schematic of the electronic structures of Ta₂NiSe₅ along the $\Gamma$–$X$-direction in the semiconductor and excitonic insulator phases proposed by previous studies[8–11]. When the exciton (which consists of an electron and a hole derived from doubly degenerate Ta $5d$ bands and the Ni $3d$-Se $4p$ band) binding energy ($E_{ex}$) exceeds the band gap ($E_{gap}$), excitons are spontaneously formed and pure Bose-Einstein condensation of excitons arises with decreasing temperature

## Results

**Pump-fluence dependence**. First, we demonstrate that Ta₂NiSe₅ (hereafter, 3% S-substituted Ta₂NiSe₅ used in this study is simply referred as Ta₂NiSe₅, since the electronic structure is almost not affected by the substitution as shown in Supplementary Fig. 2) shows a characteristic pump-fluence dependence to photo-excitation similar to another candidate of excitonic insulators, $1T$-TiSe₂. Figure 2a shows an energy–momentum ($E$–$k$) map around the $\Gamma$ point (centre of the Brillouin zone) taken before the arrival of the pump pulse at 100 K. The overall features of the

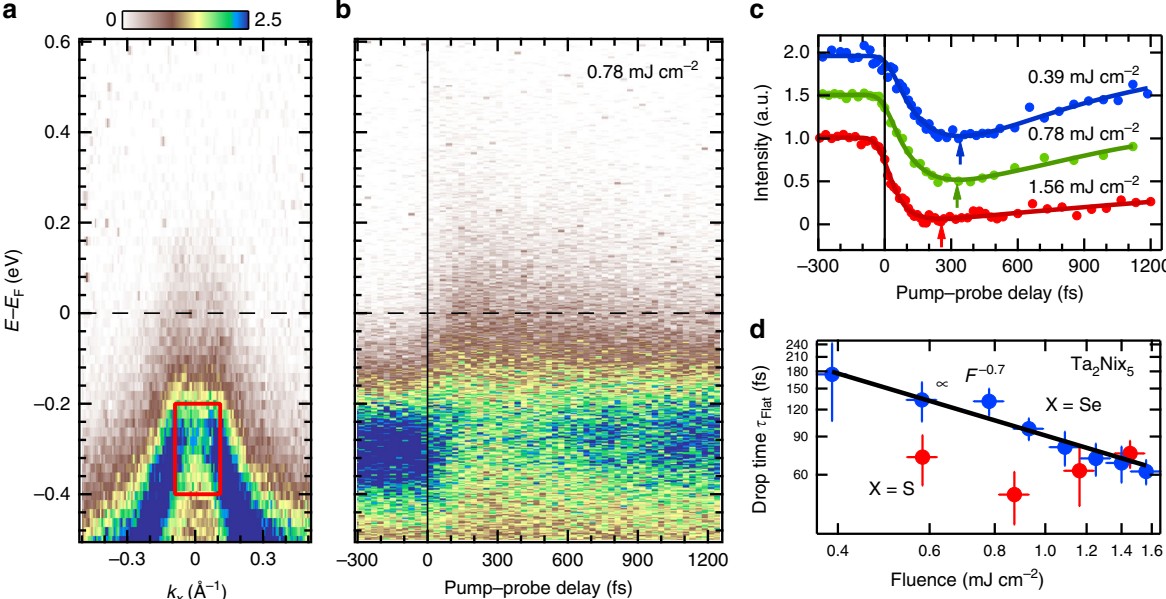

**Fig. 2** Collapse of the flat band in Ta$_2$NiSe$_5$. **a** Energy–momentum map of Ta$_2$Ni(Se$_{0.97}$S$_{0.03}$)$_5$ measured by using an XUV pulse (27.9 eV) before the arrival of the pump pulse (1.55 eV) at 100 K. **b** TARPES spectrum at the $\Gamma$ point, obtained with a pump fluence of 0.78 mJ cm$^{-2}$. **c** Temporal evolution of the integrated TARPES intensity in the red square shown in **a** for different pump fluences. The arrows indicate the minimum values of the spectral weight. **d** Extracted drop time of the flat band $\tau_{Flat}$ as a function of the pump fluence. Blue and red symbols corresponds to Ta$_2$Ni(Se$_{0.97}$S$_{0.03}$)$_5$ and Ta$_2$NiS$_5$, respectively. The error bars roughly correspond to the standard deviations

spectra are confirmed to be consistent with those of the spectrum taken at equilibrium, especially for the gap of ~250 meV, and this certifies that distortion of the spectra due to space charge effects, which could often occur for TARPES measurements with a low-repetition frequency such as 1 kHz, has been minimised. After the arrival of the pump pulse, the spectral weight of the flat band immediately decreases and is transferred to the originally gapped region at $E_F$ within 100 fs. This temporal evolution is shown in Fig. 2b. These dynamics are much faster than excess energy transfer from hot electrons to the cold phonon bath, which has been generally reported to require ~1 ps and are more likely to be associated with purely electronic process, which has been supposed to be ~100 fs or faster[18,19] In order to visualise these dynamics of the flat band after photo-excitation, in Fig. 2c we show the temporal evolution of the integrated TARPES intensity of the rectangular region in Fig. 2a as a function of the pump–probe delay ($\Delta t$) for several pump fluences. The initial decrease of the TARPES intensity strongly depends on the pump fluence and becomes faster with increasing pump fluence similar to the already reported 1$T$-TiSe$_2$ (refs. [4,20]). In order to evaluate the drop time of the flat band ($\tau_{Flat}$) (the time scale that the intensity of the flat band decreases after pumping) of Ta$_2$NiSe$_5$, the data were fitted to a Gaussian-convoluted rise-and-decay function, similar to that used in ref. [20], and the obtained values of $\tau_{Flat}$ are plotted as blue symbols in Fig. 2d.

The time scale of the gap collapse in excitonic insulators is considered to be inversely proportional to the plasma frequency, $\omega_p = (ne^2/\varepsilon_0\varepsilon_r m^\star)^{1/2}$, where $n$ is the carrier density, $e$ is the elementary charge, and $m^\star$ is the effective mass of the valence or conduction band, $\varepsilon_0$ is the electric constant, and $\varepsilon_r$ is the dielectric constant[4,20]. From this relationship, the gap quenching time should be proportional to $1/\sqrt{n}$. Since the ground state of Ta$_2$NiSe$_5$ is an insulating phase and the carrier density $n$ in the equilibrium state is expected to be quite small, the carrier density $n$ in the photo-excited state is expected to be nearly proportional to the pump fluence. We found that the drop time $\tau_{flat}$ of

Ta$_2$NiSe$_5$ was proportional to $1/F^{0.7}$, where $F$ is the pump fluence (Fig. 2d). On the other hand, we have performed the similar measurements also on Ta$_2$NiS$_5$ and the results are shown in Supplementary Fig. 3. The drop time of the top portion of the valence band was deduced from the fitting, and is plotted as red symbols in Fig. 2d. Contrastingly, it does not show clear pump fluence dependence. Thus, our results strongly suggest that Ta$_2$NiSe$_5$ is an excitonic insulator, but Ta$_2$NiS$_5$ is not (The behaviour of Ta$_2$NiS$_5$ is discussed later again). At least, the band gap of Ta$_2$NiSe$_5$ appears to originate from an electronic mechanism similar to that of 1$T$-TiSe$_2$.

**Temporal evolution of TARPES spectra.** Next, we show more impressive temporal evolution of TARPES spectra of Ta$_2$NiSe$_5$. Figure 3a, b shows the temporal evolution of the momentum-integrated energy distribution curve (EDC) and its integrated intensity above $E_F$, respectively. After the arrival of pump pulse, the intensity above $E_F$ immediately increases and relaxes with a time constant of 620 fs, which was estimated from the fitting to a Gaussian-broadened exponential decay function. Figure 3c, d shows TARPES snapshots acquired at several $\Delta t$ values and their differential spectra which were obtained by subtracting the spectrum averaged for $\Delta t < 0$, respectively (see also Supplementary Movie 1). The most notable spectral change is the emergence of an electron-like band crossing $E_F$, which is clearly seen in red colour in Fig. 3d at $\Delta t = 150$ and 250 fs. In other words, the system changes from an insulating state into a metallic state by photo-excitation. Figure 3e shows the temporal evolution of the EDCs integrated in the momentum range [−0.1, 0.1] Å$^{-1}$. Before the pump pulse arrives, the flat band appears as the strongest peak at $E - E_F$ ~−250 meV. Immediately after pumping ($\Delta t = 150$ fs), the flat band collapses and the spectral weight shifts towards higher energies. At $\Delta t = 250$ fs, the peak intensity of the flat band decreases remarkably compared to the peak at $E - E_F$ ~ −640 meV. Meanwhile, the edge at $E_F$ was found to follow the

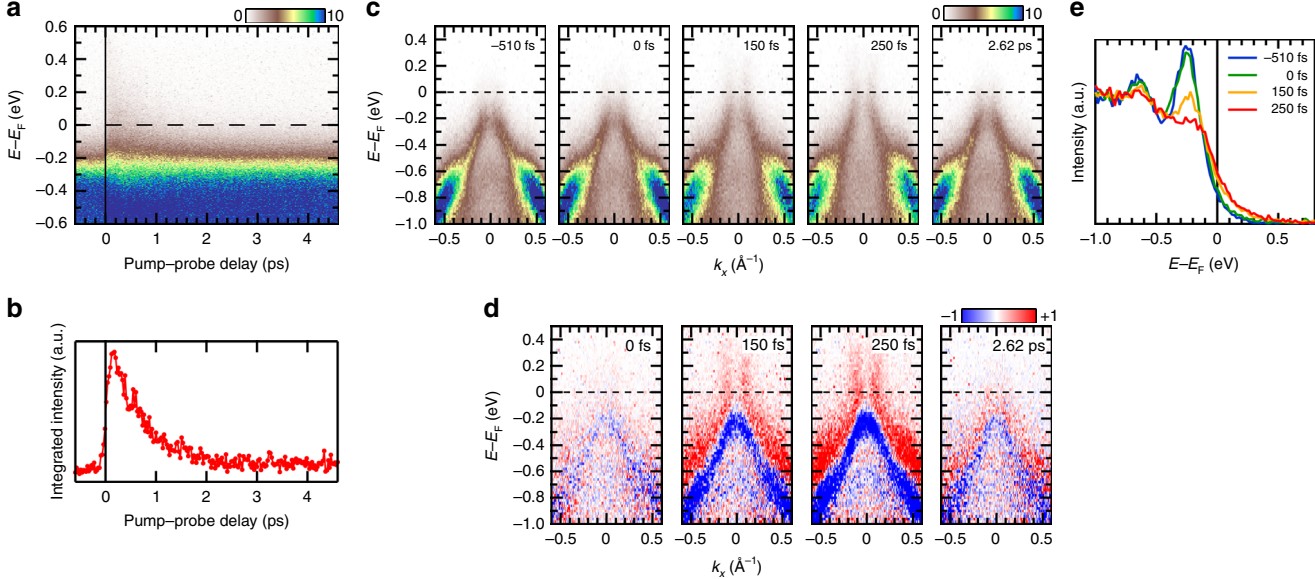

**Fig. 3** Photo-induced transition from the insulator to the metallic phase observed with TARPES measurements. **a** TARPES intensity map as a function of pump–probe delay and energy relative to $E_F$. This corresponds to the temporal evolution of the momentum-integrated EDC around the $\Gamma$ point. **b** Temporal evolution of the integrated intensity in the energy interval [0, 0.79] eV. **c** TARPES snapshots acquired at increasing pump–probe delays (see also Supplementary Movie 1). **d** Differential spectra of panel **c**, obtained by subtracting the spectrum averaged before $t_0$. **e** EDCs of the transient photoemission intensity integrated in the momentum range [−0.1, 0.1]. All time-resolved spectra were measured at 100 K with a pump fluence of 0.78 mJ cm$^{-2}$

Fermi-Dirac distribution, and the electronic temperature was estimated from the fitting to be as high as ~720 K. Since the resistivity of Ta$_2$NiSe$_5$ has been reported to become metallic-like above ~550 K[7], the observed metallic bands may correspond to this metallic behaviour at high temperature. However, this high-temperature metallic resistivity originates from thermally excited carriers and the band gap is expected to be finite even at high temperature[13]. Thus, the observed transient metallic phase could be entirely different from that observed at higher temperatures at equilibrium.

To examine the photo-induced metallic phase in more detail, we compare the time-integrated spectra before and after pumping shown in Fig. 4a, b, respectively. After photo-excitation, both the electron and hole bands cross $E_F$ at the same Fermi momentum $k_F$ ~0.1 Å$^{-1}$ as schematically shown by the red and blue parabolas in Fig. 4b [see Supplementary Figs. 4, 5 for the analysis of EDCs and momentum distribution curves before and after pumping]. This may indicate that the hybridisation between the two Ta chains is sufficiently strong to lift the degeneracy. However, since this is not predicted by band-structure calculations[10], this behaviour of the emerging of the hole and electron bands crossing $E_F$ at the same $k_F$ is a surprising nature of the observed non-equilibrium metallic phase, indicating that the observed non-equilibrium metallic state is entirely different from the high-temperature phase in the equilibrium state.

To confirm that the observed non-equilibrium metallic phase of Ta$_2$NiSe$_5$ can be associated with the excitonic condensation, we have performed comparative TARPES measurements on Ta$_2$NiS$_5$ and the results are shown in Supplementary Fig. 6. Quite unexpectedly, an electron band emerges above $E_F$ and the hole band below $E_F$ shifts upward. In addition, the bottom of the electron band and the top of the hole band seems to cross $E_F$, and the system seems likely to be semimetallic. This may require reconsidering the nature of the insulating phase for Ta$_2$NiS$_5$, which had been considered as an ordinary band insulator[12], since the valence state of nickel and tantalum is naively considered as Ni$^{0+}$(3$d^{10}$) and Ta$^{5+}$(5$d^0$).

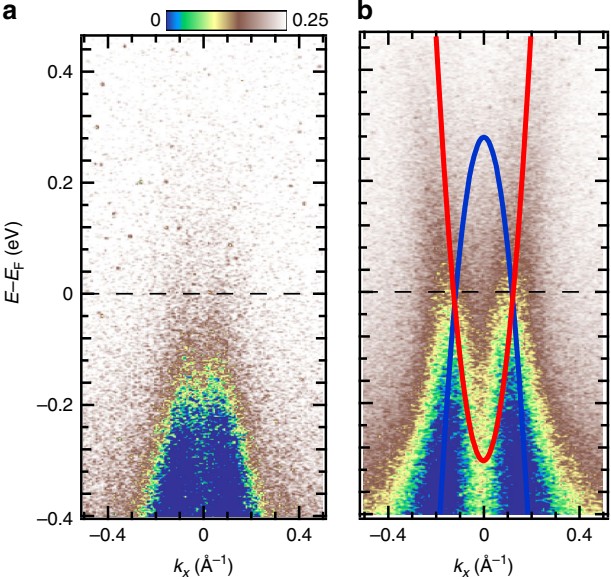

**Fig. 4** TARPES spectra of Ta$_2$NiSe$_5$ before and after pumping. **a** Energy–momentum ($E$–$k$) map before pumping, integrated in the time interval [−0.29, 0] ps. **b** Corresponding map of the transient states, integrated in [0, 1.2] ps. Red and blue parabolas indicate the electron and hole bands crossing $E_F$ in the non-equilibrium metallic state. These spectra were acquired with a pump fluence of 1.56 mJ cm$^{-2}$. Note that due to the worse energy resolution (~250 meV) compared to the static ARPES measurements shown in Supplementary Fig. 2, the $E$–$k$ map before pumping seems to have an intensity tail above $E_F$ (see also Fig. 3e)

According to the band-structure calculation based on the density functional theory[21], Ta$_2$NiS$_5$ as well as Ta$_2$NiSe$_5$ is predicted to be metallic. Hence, the electron correlation effect may be important for the origin of the band gap of Ta$_2$NiS$_5$. Wakisaka et al., has suggested this possibility for Ta$_2$NiSe$_5$,

and reported that the ground state of $Ta_2NiSe_5$ should have a significant contribution from a $d^9\underline{L}$ configuration of $Ni^{2+}$ [the valence state of tantalum is $Ta^{4+}$ ($5d^1$)], where $\underline{L}$ denotes a hole in the Se $4p$ orbital[9]. Further, $NiS_{2-x}Se_x$ is known as a bandwidth-control metal–insulator transition system, where the bandwidth is broadened by substitution of Se for S[22,23]. Similarly, the bandwidth of the $Ta_2NiS_5$ is narrower than that of $Ta_2NiSe_5$, and the electron correlation effect could be more important for the origin of the band gap of $Ta_2NiS_5$. Taking account of this possibility, the drop time of the top portion of the valence band of $Ta_2NiS_5$ can be understood. For the Mott gap, of which origin is predominantly electron correlation effect, the time scale of the gap collapse is considered to be inversely proportional to the bandwidth, and expected to be very fast[4]. Actually, the drop time of $Ta_2NiS_5$ seems to be faster than the temporal resolution of our TARPES measurements.

However, if the electronic configuration of nickel in $Ta_2NiS_5$ is close to $d^9\underline{L}$, this corresponds to that of tantalum close to $5d^1$, and mechanism for the insulating behaviour of Ta $5d$ electrons must be considered. One possibility is formation of singlet bonds between the localised Ni $3d$ spins and Ta $5d$ electrons as well as S $3p$ holes (a schematic energy diagram is shown in Supplementary Fig. 7). If one considers that Ta $5d$ electrons in the double chains form singlet states with the localised $d^9\underline{L}$ state via hybridisation with the S $3p$ orbitals, the ground state of $Ta_2NiS_5$ can be viewed as a valence-bond-like insulating state which is analogous to a Kondo insulating state. As suggested from the time scale of the gap collapse, the Mottness of the Ni $3d$ electrons increases from Se to S, and consequently, the nature of ground state is changed from the excitonic insulator to the valence-bond insulator[24].

## Discussion

Lastly, we discuss the mechanism of the photo-induced insulator-to-metal transitions realised in $Ta_2NiSe_5$ and $Ta_2NiS_5$. Whereas the dynamics of the gap collapse is governed by the interactions of the gap origin, the realisation of the non-equilibrium metallic phase cannot be understood straightforward, because no high-temperature metallic phase exists at least for $Ta_2NiS_5$. Recently, coherent phonon excitations coupled to the electronic Higgs mode has been found by the pump–probe optical measurements with the similar pump fluence to our TARPES measurements for $Ta_2NiSe_5$[14]. Also, an interesting relation between the observed coherent phonon excitations and temperature-dependent Raman spectra has been reported by Mor et al.[16] The coherent phonon excitations observed in various systems[25–27] are most likely explained by the displacive excitation of coherent phonons mechanism[28]. In this mechanism, the adiabatic energy potential is modified due to photo-excitations and has the minimum with the finite atomic displacements corresponding to the $A_g$ phonon. The electronic structure could be modulated by these lattice displacements. The observed metallic phase could be a result from this modulation of the electronic structure. If the coherent phonons of the 3.7 and/or 4 THz mode found by Mor et al. were related to the observed photo-induced phase transitions, these modes give a time scale of ~130 fs from the half-cycle time of the oscillation. This time scale could be faster for $Ta_2NiS_5$, because these modes are expected to include oscillations of Se atoms[29] and S is a lighter element than Se, and thus, regarded to be comparable to the observed gap collapse. If the observed metallic phase is driven by the modulation of the electronic structure due to the coherent lattice displacements, whereas incoherent lattice displacements driven by a large electronic density redistribution due to the strong pump pulses also could induce such modulation, as schematically shown in Fig. 1a, the observed photo-

induced transition cannot correspond to the dashed vertical arrow, but may rather correspond to the solid arrow. At least, as described above, the non-equilibrium metallic phases observed for both of $Ta_2NiSe_5$ and $Ta_2NiS_5$ should suggest that these photo-induced phase transitions are not merely transitions to higher entropy states that can be realised at high temperatures in the equilibrium state, which correspond to the dashed vertical arrow in Fig. 1a. Thus, photo-excitation can induce similar effects to pressure, and as the pressure-induced superconducting phase has been found for $Ta_2NiSe_5$, with some appropriate pumping condition probably with lower photon energy of some resonant condition, which would not give too much electronic entropy to the system, photo-induced superconductivity might be realised for this material. Realisation of this fascinating photo-induced phase would be one of the ultimate goals of investigations of the photo-excited electronic state.

## Methods

**Sample preparation**. High-quality single crystals of $Ta_2Ni(Se_{0.97}S_{0.03})_5$ and $Ta_2NiS_5$ were grown by the chemical vapour transport method. Whereas the relatively large cleaved surface is necessary for TARPES measurements compared to static ARPES, since $Ta_2NiSe_5$ has a one-dimensional crystal structure, the large cleaved surface of the pristine $Ta_2NiSe_5$ enough for TARPES measurements was difficult to obtain. However, sufficiently large cleaved surfaces of 3% S-substituted $Ta_2NiSe_5$ and $Ta_2NiS_5$ could be obtained. This is why we used 3% S-substituted $Ta_2NiSe_5$ rather than the pristine $Ta_2NiSe_5$ for this study. The results of resistivity measurements by commercial physical property measurement system (PPMS, Quantum Design) for the sample characterisation is shown in Supplementary Fig. 1. The critical temperature of the structural transition of 3% S-substituted $Ta_2NiSe_5$ is determined to be ~321 K, whereas that of the pristine $Ta_2NiSe_5$ is ~325 K. Clean surfaces were obtained by cleaving in situ.

**Photoemission measurements**. In order to characterise the cleaved surfaces and compare with the previous results, temperature-dependent static ARPES measurements were performed with a He discharge lamp and a hemispherical electron analyser (Omicron-Scienta R4000) with the energy resolution of ~12.5 meV. The results were shown in Supplementary Fig. 2. We confirmed that the temperature dependence of the top of the flat band of the 3% S-substituted sample was almost the same as the previously reported one for the pristine samples, and thus that the electronic structure was almost not affected by the 3% substitution. For the TARPES measurements, an extremely stable commercial Ti:Sapphire regenerative amplifier system (Coherent Astrella) with a centre wavelength of 800 nm ($hv$ = 1.55 eV) and pulse duration of ~30 fs was used for the pump light. After generating a second harmonic (SH) via 0.2-mm-thick β-$BaB_2O_4$, the SH light was focused into a static gas cell filled with Ar and high harmonics were generated[30]. We selected the ninth harmonic of the SH ($hv$ = 27.9 eV) for the probe light by using a set of SiC/Mg multilayer mirrors[31]. The temporal resolution was evaluated to be ~80 fs from the TARPES intensity far above the Fermi level corresponding to the cross-correlation between pump and probe pulses. The energy resolution of the spectrometer was set to ~250 meV for the TARPES measurements.

## Data availability

The data supporting the findings of this study are available from the corresponding author on request.

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

## Acknowledgements

We would like to thank H. Fukuyama and Y. Ohta for valuable discussions and comments. This work was supported by JSPS KAKENHI (Grant nos. JP25220707 and JP26610095) and the Photon and Quantum Basic Research Coordinated Development Programme from the Ministry of Education, Culture, Sports, Science and Technology, Japan. T. Someya acknowledges the JSPS Research Fellowship for Young Scientists.

## Author contributions

K.O., Y.O., T. Suzuki, T.Y., T. Someya, S.M. and M.W. performed the TARPES measurements. K.O. and Y.O. performed the data analyses. M.F., T.K., N.I. and J.I. conducted maintenance of the HHG laser system and improvements of the TARPES apparatus. Y.L., M.N., H.T., N.K. and H.S. grew high-quality single crystals and characterised them. K.O., Y.O., T. Suzuki, T.M. and S.S. wrote the manuscript. K.O., T.M. and S.S. designed the project. All authors discussed the results and contributed to the manuscript.

## Additional information

**Competing interests:** The authors declare no competing interests.

