## [Peer Review File · Nature Communications]

Reviewers' Comments:

Reviewer #1:

Remarks to the Author:

The paper reports a time- and angle-resolved XUV photoemission spectroscopy study of quasi-one-dimensional $\text{Ta}_2\text{Ni}(\text{Se}_{1-x}\text{S}_x)_5$, which exhibits a transition from a potential excitonic insulator to a potential band insulator phase upon increasing x . In particular, two compounds are investigated representing the two potentially different ground-state phases at doping levels of $x=0.03$ and $x=1$, respectively, for which transient band-structure changes are reported to occur on sub-250-fs ("sub-vibrational") timescales. The key observations are a characteristic pump-fluence dependence of the gap quenching time in the potential excitonic insulator phase, which is absent in the potential band insulator phase, as well as transient semimetallic phases in both systems. The spectroscopic results are taken as evidence that Ta_2NiSe_5 ($x=0$) in its ground state is an excitonic insulator, that the photoexcited metallic states in both Ta_2NiSe_5 and Ta_2NiS_5 ($x=1$) are distinct from high-temperature equilibrium phases, and that electronic correlation effects contribute to gap formation in Ta_2NiS_5 .

The manuscript presents a thorough, interesting, and original piece of experimental work that addresses a timely problem: the possible existence and identification of the elusive excitonic insulator in real materials. The approach chosen here, to compare dynamic spectroscopic fingerprints of isostructural compounds with similar chemical composition, is compelling. More generally, the study follows an important trend in current investigations of quantum materials, i.e., to better understand ground-state phases from the time-domain perspective. The work should therefore be of rather broad interest, not only to the ultrafast dynamics community, but also to condensed-matter physicists in general.

The reported trARPES data are significant, sufficiently novel, and well presented, the data analysis seems sound and robust, and the data interpretation appears to be reasonable. In my opinion, the manuscript is in principle appropriate for publication in Nature Communications. However, there are a number of issues that the authors need to address.

Major issues:

-- The authors should cite the recent trARPES work on Ta_2NiSe_5 by Mor et al., Phys. Rev. Lett. 119, 086401 (2017), and discuss their results in the context of that work. In particular, what are the novel aspects regarding transient band-structure changes in the present work?

-- The discussion of the transition mechanism in the final paragraph is not convincing:

First, the authors start a discussion of the potential role of coherent lattice distortions, although they do not observe any indication of coherent phonons in their data and the screening-based explanation of the electronic structure modulation is already rather conclusive. If they want to elaborate on the possible role of coherent lattice vibrations in the observed dynamics, they should at least discuss the involved timescales and their expected pump-fluence dependencies (measured transition timescales versus screening times and half-cycle times of coherent phonon oscillations).

Second, the authors make the point (lines 193 and 7) that the photo-induced electronic structure changes they detect are not "merely" due to "higher entropy". This may be so to some extent, as it indeed seems clear that the observed transient metallic states are different from high-temperature equilibrium phases. But it is also clear that using the "800-nm hammer" the authors do create a lot of electronic entropy and subsequently, upon hot electron relaxation, also phonon entropy. One could even argue that the observed screening effects are due to electronic entropy.

Third, in view of the broad non-selective excitation leading to a significant portion of hot electrons, it is difficult to imagine how a transient superconducting phase could be realized using this approach.

Minor issues:

-- The discussion of the possible ground-state electronic configurations on pages 6 and 7 (Ni and Ta oxidation states, possible Mott gap and singlet-bond formation etc.) would probably become much clearer, if it were illustrated, e.g., by some energy level schemes.

-- The softening electronic mode observed in Ref. 5 is not a "purely electronic excitation" (line 30), but possibly some hybrid plasmon-phonon mode.

-- The effective energy resolution of the trARPES measurements should be given in the Methods section.

-- The notion of a "new type of insulating state" (line 68) or "new concept of an insulating phase" (line 70) appears exaggerated.

-- The horizontal arrow illustrating the "photoexcitation" pathway in Fig. 1a is misleading. Since the excitation is not resonant with the gap, there will also be an increase in temperature, at least in the electronic system.

-- In the caption of Fig. 3a, it should be indicated that the intensity map shown corresponds to a momentum-integrated EDC taken around the Gamma point.

-- In the supplementary text, the authors refer to Figs. S2a-c, but what they probably mean is Extended Data Figs. 3a-c. More importantly, the discussion of the "increasing and decreasing rates" of spectral weight transfer is somewhat confusing. What is the point that the authors want to make here?

In conclusion, while the manuscript reports significant, novel, and potentially relevant trARPES results of an interesting material, the discussion is not conclusive and there are a number of points that need to be clarified. I therefore cannot recommend publication of this paper at this time.

Reviewer #2:

Remarks to the Author:

In this manuscript, the authors present a time- and angle-resolved photoemission spectroscopy (TRAPES) study of two samples of $\text{Ta}_2\text{Ni}(\text{Se}_{1-x}\text{S})_5$ for two different substitution values x . There is growing evidence that an excitonic insulator (EI) phase is realized in the pure semiconductor compound Ta_2NiSe_5 and recent works have demonstrated that this material displays a strong response to infrared short pulses. The EI phase occurs in Ta_2NiSe_5 at low temperature and leads to a continuous increase of its band gap as the temperature is decreased. Here, the authors show that it is possible to completely close the band gap of Ta_2NiSe_5 with infrared pump pulses of fluence $> 1.5 \text{ mJ/cm}^2$, meaning that even the original (room temperature) semiconductor band gap can be closed by the optical pulse. They also measure the intensity of the top flat valence band of Ta_2NiSe_5 as a function of time and pump fluence. By extracting the rise time of its transient intensity decrease as a function of pump fluence, they infer that the specific ultrafast dynamics confirm the presence of an EI phase in Ta_2NiSe_5 , similarly to what has been done for TiSe_2 .

Additionally, the authors study Ta₂NiS₅, which is believed to be a “normal” semiconductor with no EI phase. However, they show that its semiconductor band gap can also be closed transiently by an intense infrared pulse. They conclude the manuscript with an in-depth discussion of the photoinduced insulator-to-metal transition in both compounds.

The manuscript is well written and well structured. I find it very interesting to compare the cases of Ta₂NiSe₅, which hosts most likely an EI phase at low temperature, and of Ta₂NiS₅, which serves as a reference material. There are 2 important original results here in my opinion: the observation of a (partial) destruction of the EI phase (and its specific dynamics) and the observation of a complete collapse of the semiconductor gap. These results should deserve publication in a high impact journal as Nature Communications. However, I find that the presentation of the experimental results must be improved and some discussions should be revised, see below.

There are to date only a few time-resolved studies on Ta₂NiSe₅ and it is important to compare this new work with them. The manuscript of Werdehausen et al. is discussed here (it has been published in Science Advances 2018). However, there are a few more works (Werdehausen et al., arXiv:1801.10466, Mor et al., Phys. Rev. B 97, 115154 (2018)) and especially another TARPES study by Mor et al. (PRL 119, 086401 (2017)) that is relevant for the present manuscript. In this other TARPES study, the authors observe a very different response of Ta₂NiSe₅ to optical excitation pulses: they could not really suppress the EI phase but even observed an increase of the band gap of the material. However, excitation fluences smaller than the ones here were used. How does the current manuscript compare to these published results? Can the authors understand their different results in the light of the different excitation fluence regime used in comparison to Mor et al.? These questions must be addressed here.

The samples of Ta₂NiSe₅ used here are actually not pure compounds, but a substitution of Se by S (3%) has been done to increase the sample size. However, the results on this compound are always discussed in comparison to the pure sample. How critical (or uncritical) is this substitution for the physics of Ta₂NiSe₅? A clear characterization of this sample must be presented here and compared to the results shown in Lu et al., Nature Comm. 2017 (at least, what is the critical temperature of the phase transition in this compound? Does the position of the chemical potential help in determining if there is a substantial charge doping?).

This issue might be related to the questions raised in the previous paragraph.

The loss of intensity at the top of the valence band in Fig. 2 (photon energy 27.9 eV) is used to monitor the transient suppression of the EI phase at low temperature. Comparing Fig. 2 to Extended Data Fig. 1 (photon energy unknown?), I see that the ARPES spectra of Extended Data Fig. 1 (lower row) display the inverse behavior: the intensity in the valence band is higher around its top. Is it then correct to use the loss of intensity at the top of the valence band as a way to monitor the dynamics of the phase transition?

Would it be not better to use the intensity between this valence band and the chemical potential to do this?

Other questions and comments:

- a) The acronym TARPES is not defined in the abstract.
- b) Line 68: “Ta₂NiS₅ may be a new type of insulating state for electron-hole systems”: I find this expression quite vague. Can the authors be clearer already at the beginning of the manuscript?
- c) As far as I could see, the wavelength/energy of the pump photons is not written down. This is a very important information that must appear in the main part of the manuscript. I guess that it is 800 nm, which has been used in most works on Ta₂NiSe₅ and has been shown to lead to efficient pumping. What do the author know about the possibility of pumping Ta₂NiS₅ with 800 nm?

- d) Line 77 ff.: "the dispersions are confirmed to be consistent with those of the spectrum taken at equilibrium", which refers to a comparison between Fig. 2 and Extended Data Fig. 1. Actually, I do not find this comparison straightforward. Can the authors add fits/line to help the reader seeing this?
- e) Line 110: "the band gap of Ta₂NiSe₅ appears to originate from an electronic mechanism similar to that of 1T-TiSe₂": I find it important that the authors be precise on the nature of the band gap in Ta₂NiSe₅. From what I have read in this manuscript, I see that the low temperature band gap of Ta₂NiSe₅ has somehow 2 components, which potentially have 2 different behavior with respect to the pump pulses: the band gap (increase) due to the EI phase and the semiconductor band gap, already present at high temperatures. I think that it would globally be beneficial to the manuscript to clearly distinguish these two "components" for Ta₂NiSe₅ (in comparison to Ta₂NiS₅).
- f) In relation to Fig. 4, is it possible to observe the band gap collapse by looking at momentum distribution curves around E_f?
- g) Lines 147 ff.: in a similar way to point e) above: the authors describe the semiconductor band gap collapse only (mostly) for the case of Ta₂NiS₅. Why do they avoid including the high-temperature band gap of Ta₂NiSe₅ in this discussion?
- h) Lines 171 ff.: I do not understand the goal of this discussion. It seems to me that the authors try here to explain the unusual band gap in Ta₂NiS₅ in terms of singlet states between the Ta 5d electrons and Se 4p hole, as introduced by Wakisaka et al. (PRL 2009). However in this PRL, this singlet state has been introduced to explain the occurrence of the excitonic state in Ta₂NiSe₅. Is this not contradictory?
- i) Last paragraph of the main article: I am not convinced by the explanation of the insulator-to-metal transition with coherent phonon oscillations. Given the time resolution of 80 fs, coherent oscillations like those observed by Werdehausen et al. (who had a time resolution of 130 fs) could be observed here (although it might be difficult with the signal-to-noise ratio of the TARPES experiment). However, an incoherent lattice displacement driven by a large electronic density redistribution due to the strong pump pulses could be at the origin of the transition.
- f) Extended Data Fig. 1: what is the photon energy used here?
- g) Supplementary text, part 1: it relates to Supplementary Fig. 3 not 2.
- h) Idem: I find it confusing to speak about rates in equation (S1) and (S2), since the resulting values are dimensionless (not 1/s).

Dear Referees,

Thank you for your kind and careful review of our manuscript and giving us a lot of valuable comments and suggestions to improve our manuscript. We are very glad to know that both of you appreciate the importance of our work. In the following, we answer all the comments point by point.

Reply to the comments from Reviewer #1

-- The authors should cite the recent trARPES work on Ta₂NiSe₅ by Mor et al., Phys. Rev. Lett. 119, 086401 (2017), and discuss their results in the context of that work. In particular, what are the novel aspects regarding transient band-structure changes in the present work?

We would like to thank Reviewer #1 for this suggestion. We have cited another TARPES work by Mor *et al.*, together with other time-resolved studies on Ta₂NiSe₅ suggested from Reviewer #2, and discussed the uniqueness of our study and the reason why we obtained the different results from theirs in the revised manuscript

-- The discussion of the transition mechanism in the final paragraph is not convincing: First, the authors start a discussion of the potential role of coherent lattice distortions, although they do not observe any indication of coherent phonons in their data and the screening-based explanation of the electronic structure modulation is already rather conclusive. If they want to elaborate on the possible role of coherent lattice vibrations in the observed dynamics, they should at least discuss the involved timescales and their expected pump-fluence dependencies (measured transition timescales versus screening times and half-cycle times of coherent phonon oscillations).

As Reviewer #1 pointed out, the screening-based explanation may be rather conclusive. However, the realization of the photo-induced metallic state for Ta₂NiS₅ makes us consider that it is insufficient. Also, we have not observed any indication of coherent phonons by our TARPES measurements so far. However, whether they can be observed by TARPES or not, it should depend on how their excitations are coupled to the electronic system. Supposing that the coupling is weak, the signal-to-noise ratio could be insufficient to detect vibration signals in our measurements. On the other hand, since the coherent phonons indeed have been observed by optical measurements with the similar pumping conditions to ours, it should be reasonable to consider that they are excited by our pumping conditions, but not detected by our TARPES measurements. The pump-fluence dependences of the amplitude of the excited coherent phonons have been reported by Werdehausen

et al. If we take account of their results, since the amplitude of the coherent phonons becomes larger with the higher pump fluence (whereas the coupled mode that they found shows rather complex behaviour), the amplitude of the lattice distortions also should become larger with the higher pump fluence. Although the time scale of the transition is difficult to mention exactly because several modes of coherent excitation have been found in the range of 1-4 THz, the half-cycle times of the observed modes are in the range of 125-500 fs according to the studies by Werdehausen *et al.* and Mor *et al.* Mor *et al.* have reported interesting behaviour of the 3.7- and 4-THz modes in their static Raman and pump-probe optical measurements. We consider that whereas the dynamics of the gap collapse is governed by the interactions of the gap origin, the observed metallic phase could be a result of the coherent lattice distortions, and these modes might be responsible for the distortions. The related discussions have been added to the last paragraph of the revised manuscript.

Second, the authors make the point (lines 193 and 7) that the photo-induced electronic structure changes they detect are not “merely” due to “higher entropy”. This may be so to some extent, as it indeed seems clear that the observed transient metallic states are different from high-temperature equilibrium phases. But it is also clear that using the “800-nm hammer” the authors do create a lot of electronic entropy and subsequently, upon hot electron relaxation, also phonon entropy. One could even argue that the observed screening effects are due to electronic entropy.

We agree that using the “800-nm hammer”, a lot of electronic entropy would be created. However, since there exists no temperature-induced phase transition for Ta₂NiS₅ at least up to 550 K, this fact makes us conclude that the screening effects due to electronic entropy should be insufficient for the realization of the photo-induced metallic phase of Ta₂NiS₅.

Third, in view of the broad non-selective excitation leading to a significant portion of hot electrons, it is difficult to imagine how a transient superconducting phase could be realized using this approach.

We agree that it should be difficult to realize a transient superconducting state using the pump photons of the “800-nm hammer”. On the other hand, if the observed photo-induced transition was actually realized by displacive excitations, and such displacive excitations were possible with lower photon energy of some resonant condition, it could be expected that creation of hot electrons is limited to the minimum. In this meaning, we used a term of “with some appropriate condition”. We have added related discussions in the last paragraph of the revised manuscript.

Minor issues:

-- The discussion of the possible ground-state electronic configurations on pages 6 and 7 (Ni and Ta oxidation states, possible Mott gap and singlet-bond formation etc.) would probably become much clearer, if it were illustrated, e.g., by some energy level schemes.

As Reviewer #1 suggested, we have made a schematic energy diagram in Supplementary Fig. 7, and also polished the related discussions.

-- The softening electronic mode observed in Ref. 5 is not a “purely electronic excitation” (line 30), but possibly some hybrid plasmon-phonon mode.

As Reviewer #1 pointed out, “purely electronic excitation” is not a valid description for here. We have changed the expression here to “electronic collective modes coupled to phonons”.

-- The effective energy resolution of the trARPES measurements should be given in the Methods section.

The energy resolution of the TARPES measurements was 250 meV. This has been given in the Methods section, as Reviewer #1 suggested.

-- The notion of a “new type of insulating state” (line 68) or “new concept of an insulating phase” (line 70) appears exaggerated.

According to this comment from Reviewer #1, we removed these exaggerated expressions and revised the manuscript accordingly.

-- The horizontal arrow illustrating the “photoexcitation” pathway in Fig. 1a is misleading. Since the excitation is not resonant with the gap, there will also be an increase in temperature, at least in the electronic system.

In Fig. 1a, we would like to discuss whether the observed photo-induced phase transition is “merely” due to effect of increasing temperature by pump or not. Since Ta_2NiS_5 has no temperature-induced phase transition, and its high-temperature metallic phase cannot exist, the transition should be different from that “merely” due to effect of increasing temperature, and the gap should be intrinsically controlled by pump. In this meaning, we have considered that the pathway for the

observed transition is rather a horizontal arrow than a vertical arrow. However, as Reviewer #1 pointed out, it might be misleading. We have changed this arrow to the upper left direction to include some amount of the effect of increasing temperature.

-- In the caption of Fig. 3a, it should be indicated that the intensity map shown corresponds to a momentum-integrated EDC taken around the Gamma point.

As Reviewer #1 suggested, we have indicated this point in the caption of Fig. 3a.

-- In the supplementary text, the authors refer to Figs. S2a-c, but what they probably mean is Extended Data Figs. 3a-c. More importantly, the discussion of the “increasing and decreasing rates” of spectral weight transfer is somewhat confusing. What is the point that the authors want to make here?

As Reviewer #1 pointed out, the reference to Figs. S2 is our mistake, and the correct one is Extended Data Fig. 3 (Supplementary Fig. 4 in the revised manuscript). In addition, the word “rate” here might be confusing, as Reviewer #2 also pointed out. We re-define the quantities described in Eqs. S1 and S2 as “fraction of increase” and “fraction of decrease”, respectively. In order to visualize the pump-induced spectral variation more clearly, we have shown the EDCs at various momenta in Supplementary Fig. 4a. From these EDCs, we can see that intensity of the EDCs after pump decreases around the region around the flat band top ($\omega_1 < E - E_F < \omega_2$), and increases above it ($E - E_F > \omega_2$). These variations at various momenta roughly correspond to the variation of the MDCs at the top of the flat band and around E_F . In addition, we show the MDCs with the energy integration window of ± 0.1 eV, of which signal-to-noise ratio is not so good, in Supplementary Fig. 5. Basically, the MDC around the top of the flat band before pump has an intensity maximum around the zone center, whereas the peak positions of the MDC around E_F after pump correspond to the k_F positions.

Reply to the comments from Reviewer #2

There are to date only a few time-resolved studies on Ta₂NiSe₅ and it is important to compare this new work with them. The manuscript of Werdehausen et al. is discussed here (it has been published in Science Advances 2018). However, there are a few more works (Werdehausen et al., arXiv:1801.10466, Mor et al., Phys. Rev. B 97, 115154 (2018)) and especially another TARPES study by Mor et al. (PRL 119, 086401 (2017)) that is relevant for the present manuscript. In this other TARPES study, the authors observe a very different response of Ta₂NiSe₅ to optical excitation pulses: they could not really suppress the EI phase but even

observed an increase of the band gap of the material. However, excitation fluences smaller than the ones here were used.

How does the current manuscript compare to these published results? Can the authors understand their different results in the light of the different excitation fluence regime used in comparison to Mor et al.? These questions must be addressed here.

As Reviewer #2 pointed out, we consider that the comparison to the other time-resolved studies are extremely important to fully understand our results. We have cited the references suggested from Reviewer #2, and discussed the origins of the different results from Mor *et al.* in the revised manuscript. As Reviewer #2 pointed out, the different excitation fluence is supposed to be one of them. In addition, the better temporal resolution of our measurements may be another. Whereas they estimated the upper limit for the time resolution of 110 fs, we employed a laser system with a shorter pulse duration of 30 fs and estimated the upper limit for the time resolution of 80 fs.

The samples of Ta₂NiSe₅ used here are actually not pure compounds, but a substitution of Se by S (3%) has been done to increase the sample size. However, the results on this compound are always discussed in comparison to the pure sample. How critical (or uncritical) is this substitution for the physics of Ta₂NiSe₅? A clear characterization of this sample must be presented here and compared to the results shown in Lu et al., Nature Comm. 2017 (at least, what is the critical temperature of the phase transition in this compound? Does the position of the chemical potential help in determining if there is a substantial charge doping?).

This issue might be related to the questions raised in the previous paragraph.

We have compared the conventional static ARPES spectra of our 3 % S substituted samples to those of the pristine sample measured by Wakisaka *et al.*, and then concluded that there is no significant difference between these two compounds. In addition to this fact, we have added the results of resistivity measurements of the substituted sample compared to the pristine one as Supplementary Fig. 1 for more detailed sample characterization. The critical temperature is slightly different by ~4 K, but the activation energy ($E_p = -k_B T^2 (\partial \ln \rho / \partial T)$) is almost the same below and above the critical temperature.

The loss of intensity at the top of the valence band in Fig. 2 (photon energy 27.9 eV) is used to monitor the transient suppression of the EI phase at low temperature. Comparing Fig. 2 to Extended Data Fig. 1 (photon energy unknown?), I see that the ARPES spectra of Extended Data Fig. 1 (lower row) display the inverse behavior: the intensity in the valence band is higher around its top. Is it then correct to use the loss of intensity at the top of the valence band as a

way to monitor the dynamics of the phase transition?

Would it be not better to use the intensity between this valence band and the chemical potential to do this?

The spectra shown in Extended Data Fig. 1 (Supplementary Fig. 2 in the revised manuscript) were taken with He I α resonance line and the photon energy was 21.2 eV. The existence of the flat band at the top of the valence band has been considered to be the most indicative feature for the excitonic phase of this material. The spectra shown in Supplementary Fig. 2 were taken in the excitonic phase, and thus the intensity in the valence band is higher around its top. We have considered that the reduction of the intensity around this region could be a measure for the collapse of the excitonic gap. The similar analysis has been performed in Fig. 2 of Ref. 20

Other questions and comments:

a) The acronym TARPES is not defined in the abstract.

We have avoided to use the acronym TARPES in the abstract of the revised manuscript.

b) Line 68: “Ta₂NiS₅ may be a new type of insulating state for electron-hole systems”: I find this expression quite vague. Can the authors be clearer already at the beginning of the manuscript?

Reviewer #1 also pointed out that this expression is exaggerated. We have revised the manuscript related to this point.

c) As far as I could see, the wavelength/energy of the pump photons is not written down. This is a very important information that must appear in the main part of the manuscript. I guess that it is 800 nm, which has been used in most works on Ta₂NiSe₅ and has been shown to lead to efficient pumping. What do the author know about the possibility of pumping Ta₂NiS₅ with 800 nm?

We agree that the wavelength of the pump photon is a very important information. While we had described the wavelength of the pump laser in the Methods section, we have specified its photon energy in the caption of Fig. 2 as well as the Methods section. The optical conductivity measurements for Ta₂NiSe₅ and Ta₂NiS₅ have been reported by Larkin *et al.* (Ref. 21), and according to this result, Ta₂NiS₅ may be excited more efficiently.

d) Line 77 ff.: “the dispersions are confirmed to be consistent with those of the spectrum taken at equilibrium”, which refers to a comparison between Fig. 2 and Extended Data Fig. 1. Actually, I do not find this comparison straightforward. Can the authors add fits/line to help the reader seeing this?

As Reviewer #2 pointed out, we agree that a comparison of the dispersions between Fig. 2 and Extended Data Fig. 1 (Supplementary Fig. 2 in the revised manuscript) is not straightforward. Here, the meaning that “the dispersions are confirmed to be consistent” simply indicated that the gap between the valence band top and E_F is ~ 250 meV at 100 K for both spectra. We have specified it in the revised manuscript.

e) Line 110: “the band gap of Ta₂NiSe₅ appears to originate from an electronic mechanism similar to that of 1T-TiSe₂”: I find it important that the authors be precise on the nature of the band gap in Ta₂NiSe₅. From what I have read in this manuscript, I see that the low temperature band gap of Ta₂NiSe₅ has somehow 2 components, which potentially have 2 different behavior with respect to the pump pulses: the band gap (increase) due to the EI phase and the semiconductor band gap, already present at high temperatures. I think that it would globally be beneficial to the manuscript to clearly distinguish these two “components” for Ta₂NiSe₅ (in comparison to Ta₂NiS₅).

As Reviewer #2 pointed out, the band gap of Ta₂NiSe₅ might have two components, since Ta₂Ni(Se,S)₅ had been considered that they are located at the BEC side in the phase diagram of excitonic insulators shown in Fig. 1. However, we consider that it is extremely difficult to experimentally distinguish the two components, since the gap of Ta₂NiSe₅ totally collapses after optical pumping, whereas we had expected that only excitonic gap is reduced by screening effect of photo-excited carriers. In addition, since now we found out that the gap of Ta₂NiS₅, which had been considered as an ordinary band insulator, also collapse by photo-excitations, we consider that we cannot distinguish the two components any more.

f) In relation to Fig. 4, is it possible to observe the band gap collapse by looking at momentum distribution curves around E_f ?

Generally, TARPES measurements are quite hard to obtain spectra with a good signal-to-noise ratio, typically with the repetition rate of the laser as low as 1 kHz. Whereas we have shown the momentum integrated EDCs in Fig. 3e, the momentum integration range is as wide as $[-0.1, 0.1] \text{ \AA}^{-1}$.

In order to settle this difficulty somehow, we defined the “increasing rate” and “decreasing rate”, which now we have redefined as “fraction of increase” and “fraction of decrease”, respectively, and plotted them for various momenta. These should approximately correspond to the MDCs around E_F after pump and around the flat band top before pump, respectively. In addition to this, we have plotted the MDCs around E_F and the flat band top with the energy integration window of ± 0.1 eV.

g) Lines 147 ff.: in a similar way to point e) above: the authors describe the semiconductor band gap collapse only (mostly) for the case of Ta₂NiS₅. Why do they avoid including the high-temperature band gap of Ta₂NiSe₅ in this discussion?

As we mentioned in the reply to the comment e) from Reviewer #2, since we could not experimentally confirm the existence of two components of the band gap, the excitonic gap and the semiconductor gap, for Ta₂NiSe₅, and considered that we could not distinguish them, we have not included the high-temperature band gap of Ta₂NiSe₅ in this discussion.

h) Lines 171 ff.: I do not understand the goal of this discussion. It seems to me that the authors try here to explain the unusual band gap in Ta₂NiS₅ in terms of singlet states between the Ta 5d electrons and Se 4p hole, as introduced by Wakisaka et al. (PRL 2009). However in this PRL, this singlet state has been introduced to explain the occurrence of the excitonic state in Ta₂NiSe₅. Is this not contradictory?

S is an isovalent element with Se. Hence, the possible electronic configuration is similar to each other. According to the discussions by Wakisaka *et al.*, creation of singlet states may be common for both of Ta₂NiSe₅ and Ta₂NiS₅. However, we have considered that the Mottness of Ni 3d electrons increases with S substitution for Se, and as the result, Ta₂NiS₅ can be regarded as a valence-bond insulator, while Ta₂NiSe₅ is an excitonic insulator.

i) Last paragraph of the main article: I am not convinced by the explanation of the insulator-to-metal transition with coherent phonon oscillations. Given the time resolution of 80 fs, coherent oscillations like those observed by Werdehausen et al. (who had a time resolution of 130 fs) could be observed here (although it might be difficult with the signal-to-noise ratio of the TARPES experiment). However, an incoherent lattice displacement driven by a large electronic density redistribution due to the strong pump pulses could be at the origin of the transition.

This comment seems to be related to the comment from Reviewer #1. One of the reasons why we

have not been able to detect coherent phonon excitations by our TARPES measurements would be the insufficient signal-to-noise ratio, and the weak coupling of the phonons to the valence-band electrons could be another. However, as written in the reply to that comment, since the pumping conditions for our TARPES measurements is similar to those of Werdehausen *et al.*, it should be reasonable that the coherent phonons are excited with our pumping conditions. Based on the experimental fact that Ta₂NiS₅, which had been considered as an ordinary band insulator, also shows a photo-induced phase transition, the screening effect only should be insufficient for the mechanism of the transitions, and then we considered another mechanism is necessary to be considered. Anyway, we agree that, with our pumping condition of 1.55 eV phonons, an incoherent lattice displacement driven by a large electronic density redistribution could be another reason. We have mentioned this possibility in the revised manuscript.

j) Extended Data Fig. 1: what is the photon energy used here?

As we described above, the spectra in Extended Data Fig. 1 were taken with He I α resonance line, and the photon energy was 21.2 eV.

k) Supplementary text, part 1: it relates to Supplementary Fig. 3 not 2.

As Reviewer #2 pointed out, Supplementary text, part 1 is related to Extended Data Fig. 3 (Supplementary Fig. 4 in the revised manuscript). We have corrected this mistake.

l) Idem: I find it confusing to speak about rates in equation (S1) and (S2), since the resulting values are dimensionless (not 1/s).

As we have mentioned above, we re-define the quantities described in Eqs. S1 and S2 as “fraction of increase” and “fraction of decrease”, respectively, since the word “rate” here might be confusing.

We hope both of you find our revised manuscript suitable for publication in *Nature Communications*.

Sincerely yours,

Reviewer #1:

Remarks to the Author:

The authors have addressed all of the reviewers' points. Most of the responses are convincing, and the changes made to the main manuscript and supplementary material have improved the overall clarity and quality of the work. The results are now also placed in a context of previous time-resolved studies of the same class of materials, although one could certainly have expected more specific statements about what has actually been observed in those studies (in particular in Ref. 15) regarding ultrafast electronic structure changes. While another round of review will probably not lead to further significant improvements, I would nevertheless recommend a careful proofreading of the manuscript and supplementary material. In conclusion, I think that the response is satisfactory and that the paper can be published in Nature Communications.

Reviewer #2:

Remarks to the Author:

In this new version of their manuscript, the authors have made efforts to improve the readability of the main text and have also augmented the supplementary material to support further their conclusions. They have also better referenced their work with respect to the recently published literature.

I am however still disturbed by two points that I have already raised in my first report and that should be better addressed, before I can definitively recommend this manuscript for publication in Nature Communications.

The samples of Ta₂NiSe₅ used here are obtained by substitution of Se by S (3%). To confirm that this small substitution has little influence on the physics of Ta₂NiSe₅, the authors have added Supplementary Figure 1 displaying the resistivity curves of pure and substituted samples, showing that 3% of S results in a shift of only 4 K of the transition temperature. However, in Supplementary Figure 2, they show ARPES spectra of these samples indicating a position of the valence band at Gamma of about 200 meV at 300 K, which is significantly larger than what is obtained in Ref. 9 (about 120 meV at 300 K). How do they explain such a difference? This indicates to electron-doped samples. Such a difference with a reference work in the literature is important to note in the manuscript and might have some influence on the excited electron and hole dynamics.

My last criticism is still about the interpretation involving the excitation of coherent phonons, in the last paragraph. In the new version, the authors write that the half-cycle of the 3.7 and/or 4 THz phonon modes is about 130 fs and might even be faster for Ta₂NiS₅ (although I do not really agree with the argument, since it is speculative to attribute these phonon modes to Se/S atoms). I understand that it might be hard to observe them in such a TARPES experiment, given the signal-to-noise ratio achieved here. However, if the photoinduced metallic phase would be the result of such a coherent phonon oscillation, I have a hard time to figure out how this could be compatible with the decay time of the states above E_f shown in Fig. 3b, since it shows a typical incoherent response with a decay time much longer than 130 fs.

One can argue that it reflects not only the band gap collapse, but also the electrons excited in the conduction band. In that case, the authors should display similar data integrated just around E_f to support their discussion.

Dear Referees,

Thank you again for your kind and careful review of our manuscript. We are very glad to know that Reviewer #1 recommends our manuscript to be published in Nature Communications, and also that Reviewer #2 appreciates our efforts to improve our manuscript. In the following, we answer the remaining concerns of both referees.

Reply to the comments from Reviewer #1

The results are now also placed in a context of previous time-resolved studies of the same class of materials, although one could certainly have expected more specific statements about what has actually been observed in those studies (in particular in Ref. 15) regarding ultrafast electronic structure changes.

More specifically, Mor *et al.* has observed the band gap narrowing and enhanced excitonic coupling with the pump fluences ranging from 0.03 to 0.47 mJ/cm², which are lower than those of our measurements ranging from 0.39 to 1.56 mJ/cm². The related statement has been added in the manuscript.

While another round of review will probably not lead to further significant improvements, I would nevertheless recommend a careful proofreading of the manuscript and supplementary material.

We have carefully checked the manuscript and supplementary material, and corrected several minor mistakes before resubmission.

Reply to the comments from Reviewer #2

The samples of Ta₂NiSe₅ used here are obtained by substitution of Se by S (3%). To confirm that this small substitution has little influence on the physics of Ta₂NiSe₅, the authors have added Supplementary Figure 1 displaying the resistivity curves of pure and substituted samples, showing that 3% of S results in a shift of only 4 K of the transition temperature. However, in Supplementary Figure 2, they show ARPES spectra of these samples indicating a position of the valence band at Gamma of about 200 meV at 300 K, which is significantly larger than what is obtained in Ref. 9 (about 120 meV at 300 K). How do they explain such a

Reviewers' Comments:

Reviewer #2:

Remarks to the Author:

The authors have answered satisfactorily to my questions. I recommend now publication in Nature Communications.